# T cell stimulation remodels the latently HIV-1 infected cell population by differential activation of proviral chromatin

**Birgitta Lindqvist**[1]☯, **Bianca B. Jütte**[1]☯, **Luca Love**[1]☯, **Wlaa Assi**[1,2], **Julie Roux**[1], **Anders Sönnerborg**[3], **Tugsan Tezil**[4], **Eric Verdin**[4], **J. Peter Svensson**[1]*

**1** Department of Biosciences and Nutrition, Karolinska Institutet, Huddinge, Sweden, **2** Laboratory of Viral Infectious Diseases, Department of Computational Biology and Medical Sciences, Graduate School of Frontier Sciences, The University of Tokyo, Tokyo, Japan, **3** Division of Clinical Microbiology, Department of Laboratory Medicine, Karolinska Institutet, Stockholm, Sweden, Division of Infectious Diseases, Department of Medicine Huddinge, I73, Karolinska University Hospital, Stockholm, Sweden, **4** Buck Institute for Research on Aging, Novato, California, United States of America

☯ These authors contributed equally to this work.
* peter.svensson@ki.se

**Data Availability Statement:** The ChIP-seq data have been deposited in the GEO database under ID GSE183275. Published dataset from (Reeder et al., 2015) [36] was retrieved from GEO at GSE65689.

## Abstract

The reservoir of latently HIV-1 infected cells is heterogeneous. To achieve an HIV-1 cure, the reservoir of activatable proviruses must be eliminated while permanently silenced proviruses may be tolerated. We have developed a method to assess the proviral nuclear microenvironment in single cells. In latently HIV-1 infected cells, a zinc finger protein tethered to the HIV-1 promoter produced a fluorescent signal as a protein of interest came in its proximity, such as the viral transactivator Tat when recruited to the nascent RNA. Tat is essential for viral replication. In these cells we assessed the proviral activation and chromatin composition. By linking Tat recruitment to proviral activity, we dissected the mechanisms of HIV-1 latency reversal and the consequences of HIV-1 production. A pulse of promoter-associated Tat was identified that contrasted to the continuous production of viral proteins. As expected, promoter H3K4me3 led to substantial expression of the provirus following T cell stimulation. However, the activation-induced cell cycle arrest and death led to a surviving cell fraction with proviruses encapsulated in repressive chromatin. Further, this cellular model was used to reveal mechanisms of action of small molecules. In a proof-of-concept study we determined the effect of modifying enhancer chromatin on HIV-1 latency reversal. Only proviruses resembling active enhancers, associated with H3K4me1 and H3K27ac and subsequentially recognized by BRD4, efficiently recruited Tat upon cell stimulation. Tat-independent HIV-1 latency reversal of unknown significance still occurred. We present a method for single cell assessment of the microenvironment of the latent HIV-1 proviruses, used here to reveal how T cell stimulation modulates the proviral activity and how the subsequent fate of the infected cell depends on the chromatin context.

**Funding:** JPS was supported by grant 2019-00991
Vetenskapsrådet (www.vr.se); and 12 0412 Pj
Cancerfonden (www.cancerfonden.se); and
Fob2020-0004 Stiftelsen Läkare mot AIDS
Forskningsfond (http://www.aidsfond.se/). JPS and
AS were supported by grant FoUI-954473 Center
for Innovative Medicine (https://cimed.ki.se/). The
funders had no role in study design, data collection
and analysis, decision to publish, or preparation of
the manuscript.

**Competing interests:** The authors declare that they
have no conflict of interest.

## Author summary

Upon infection with HIV-1, viral DNA is integrated into the cellular genome and adopts
the surrounding chromatin structure. Integrated viral DNA can lead to production of new
viral particles, or remain latently integrated in the cell. The chromatin environment
strongly influences this state. Moreover, the viral protein Tat plays an important role in
reversing latency and spreading the infection. In this work, we have developed a method
to assess the microenvironment of the integrated HIV-1 promoter in single cells.
Although continuous viral protein production occurred after T cell stimulation, we found
that Tat was present only briefly at the viral promoter and the chromatin landscape
around the HIV-1 locus was altered. This new method helps to understand the impact of
specific chromatin modifications on the HIV-1 activation potential. Using small mole-
cules, we also modified chromatin marks and analyzed their effect on HIV-1 latency.
With this study, we provide new insights in the cellular mechanisms of HIV-1 latency and
its reversal, which sheds light on the struggles to overcome to achieve a functional HIV-1
cure.

## Introduction

Despite antiretroviral therapy (ART) potently inhibiting HIV-1 replication, the intact HIV-1
genome persists in infected cells. Most of these cells do not produce viral particles and hence
make up the latent reservoir. Some proviral transcription is observed in these cells but rarely
are the transcripts translated [1], thereby the cells evade immune recognition. Efforts to stimu-
late the cells and expose them as infected to the immune system has failed clinically [2–4] as
even the strongest activators lead to limited HIV-1 latency reversal [5,6]. Among people living
with HIV-1, rare individuals naturally control viral replication, a control driven by a potent
immune response that kill cells exposing viral proteins [7] in combination with a majority of
proviruses integrated in repressive chromatin [8]. This combined approach of eliminating the
reservoir of activatable proviruses while tolerating permanently silenced proviruses, produces
a sustained, drug-free HIV-1 remission–a *de facto* HIV-1 cure [9].

   To reproduce this HIV-1 remission clinically in a more general population, we must target
each sub-compartment of the latent reservoir specifically. As discrete sub-compartments may
be under specific control, first we need to delineate the distinct mechanisms governing HIV-1
latency reversal [10,11]. A well-studied reservoir fraction contains proviruses integrated in
active regions. These open chromatin structures contain most proviruses upon initial integra-
tion [5,12,13]. As the HIV-1 capsid enters the nucleus through the nuclear pore [14], the provi-
rus almost immediately integrates, usually within a short distance [15,16]. The
microenvironment of the nuclear pore is a highly transcriptionally permissive location [17].
Stochastic reversal of proviral latency is induced by T cell activation. Upon activation, the
gene-like structure of HIV-1, with H3K4me3 at the promoter and a splice donor immediately
downstream the transcription start site [18,19], allows mRNA transcription, processing and
translation of viral proteins, including the essential transactivator of transcription Tat [1].

   Another reservoir sub-compartment is contained in enhancer-like structures–defined as
H3K4me1 and H3K27ac chromatin and associated with short unspliced transcripts–where
proviruses remain accessible and with high reactivation potential [1,5,13,20,21]. Enhancer
chromatin structures enable long-term reactivatable latency as it provides an open chromatin
without mRNA production [21]. In contrast, the initially rare cellular fractions where proviral
genomes are embedded in heterochromatin marks H3K9me3 or H3K27me3 have low

reactivation potential [5,22,23]. H3K9me3 contained proviruses may still be activated by knock-down of heterochromatin protein 1 (HP1) [22] but their clinical contribution is uncertain. Further, the HIV-1 reservoir is dynamic and evolves over time [24–26]. Proviruses in heterochromatin gradually become more prominent among the HIV-1 infected cells, at the expense of proviruses with components of actively transcribed regions [21,27]. This is caused by a combination of clonal expansion of cells with silenced provirus, elimination of actively HIV-1 producing cells and through chromatin of individual proviruses transitioning from active to inactive. Heterochromatin forms over the HIV-1 promoter in the absence of functional Tat [28].

For *in vivo* replication of the HIV-1, the Tat protein is essential. Although Tat-independent transcription account for the initial proviral transcription in latently infected cells, in the absence of Tat HIV-1 transcription initiation is normal but only produces short abortive transcripts [29]. In an early step of HIV-1 latency reversal, Tat is recruited to the long terminal repeat (LTR) promoter to override the host-controlled transcription machinery and augment proviral expression [30]. Tat binds to an initial nucleotide sequence of the nascent HIV-1 RNA, the trans-activator response (TAR) element. It then recruits and modifies the positive transcription elongation factor b (P-TEFb) complex to enable transcription elongation. Once host mechanisms are assembled for transcription, Tat is acetylated and loses its affinity for RNA [31,32]. In the viral circuitry, Tat has been proposed to generate promoter toggling, *i.e.* being responsible for both the ON and OFF switch of the HIV-1 provirus [33,34]. Except for regulating the HIV-1 promoter, ectopic expression of Tat has been observed to have affinity for other genomic regions [35,36]. Apart from promoting transcription, Tat stimulates activation induced cell death (AICD) [37]. AICD is a naturally occurring process where activated T cells induce apoptosis after an immune response as to maintain the balance of T cells once infection is cleared. In this process, the cells progress through the cell cycle and arrest in G1 [38]. As cell stress and death in itself induce HIV-1 reactivation [39] and cells with activated HIV-1 express toxic viral proteins that further reduce the viability of virus, this generates a positive feed-back loop of HIV-1 latency reversal and cell death. This process becomes particularly strong in cellular models. In models with a late readout such as the accumulated production of p24 (or a reporter protein such as GFP), HIV-1 latency reversal induced by T cell activation or as a consequence of AICD, become inseparable, making it impossible to dissect the mechanisms of latency reversal.

Here we present a single cell method to assess chromatin at the HIV-1 locus and link it to early stages of HIV-1 activation. Using this method, we show that upon T cell stimulation, Tat was transiently recruited to the HIV-1 promoter in proliferating cells. Following T cell stimulation, the proviral chromatin landscape within the HIV-1 infected cells was altered due to selective cell death. We also found that Tat was uniquely present at the HIV-1 promoter when it had enhancer features delivered by CBP/P300 and BRD4.

## Results

### Labeling of the HIV-1 provirus to enable proximity ligation assay at the promoter

To study the microenvironment of the HIV-1 promoter in single cells, we adapted the well-established proximity ligation assay (PLA) [40]. Performing PLA in fixed cells results in a bright fluorescent signal when two query proteins are within 40 nm of each other (Fig 1A). PLA relies on the proximity of two antibodies that have been conjugated with an oligonucleotide probe. To enable detection of the integrated HIV-1, we tethered a small protein to the proviral promoter region. Zinc finger proteins (ZFP) bind to DNA sequences in a highly specific

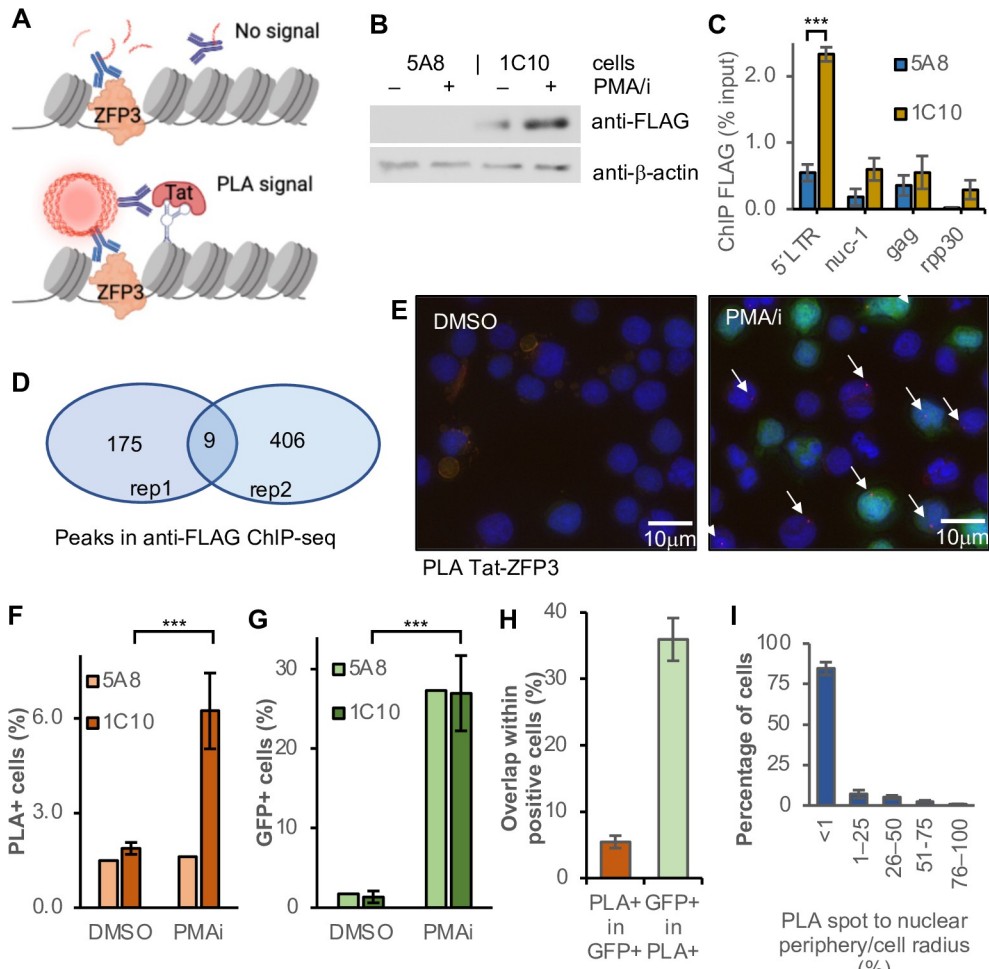

**Fig 1. PLA detects Tat at the HIV-1 promoter.** (A) The HIV-1 promoter with the zinc finger protein ZFP3 (orange), and an adjacent protein, here Tat (pink) recognized by two oligonucleotide-conjugated antibodies that are ligated to enable rolling circle amplification, and subsequent binding of fluorescently labelled probes during PLA. (B) Immunoblot using antibody against FLAG in 1C10 cells, a clone of HIV-1 containing J-lat 5A8 cells infected with FLAG-tagged ZFP3, and parental 5A8 cells treated with DMSO or PMA/i for 24 h. (C) ChIP-qPCR of FLAG in 1C10 and 5A8 cells ($n = 3$, s.e.m.). (D) Venn diagram of MACS2 peaks (filtered: -log(q)>5) from ChIP-seq with anti-FLAG in 1C10 cells ($n = 2$). (E) PLA with 1C10 cells treated with DMSO or PMA/i for 16 h. Nuclei were stained with DAPI, green shows GFP expression and red shows PLA spots. White arrows points to the PLA spots. (F–G) Quantification of Tat-ZFP3 PLA$^+$ cells (F) and GFP$^+$ cells (G), (H) The overlap between GFP$^+$ and Tat-ZFP3 PLA$^+$ cells. (I) Distance of PLA spots to nuclear periphery ($n = 5$). (B, F, G) Student t-test p-values $^*$ p<0.05, $^{**}$ p<0.01, $^{***}$p<0.005.

manner. A specific protein, ZFP3, has been artificially constructed to recognize the HIV-1 promoter, adjacent to the NFκB binding sites [41]. We cloned a FLAG-labelled ZFP3 protein into a lentiviral plasmid under the SFFV promoter. The ZFP3 sequence was linked to the BFP sequence through the sequence for a self-cleaving P2A peptide. Lentiviral particles were transduced into J-lat 5A8 cells, a Jurkat cell line harboring a single copy of latent HIV-1 [42,43]. The latent reporter HIV-1 in 5A8 cells is a full-length mutated provirus where *env* has a frameshift mutation and *nef* is replaced by a GFP coding sequence. After clonal expansion of ZFP3-BFP transduced cells, for further studies we selected a clone, 1C10, with 96.4% ZFP3 positive cells, as identified by BFP expression (S1A Fig). The FLAG-ZFP3 protein was present in 1C10 cells in both unstimulated cells and cells stimulated with phorbol 12-myristate 13-acetate and ionomycin (PMA/i) as confirmed by immunoblotting (Fig 1B). Cell stimulation with

PMA/i leads to highly correlated transcription of viral RNA and GFP expression in these cells [21,42,44]

## The engineered zinc-finger protein ZFP3 binds to the HIV-1 promoter

Next, we tested the binding specificity of the engineered ZFP3 to its target DNA sequence. The ZFP3 recognizes a 14 nt sequence found in the proviral LTR (position 408–422 at the 5′LTR and 9,493–9,507 at the 3′LTR in the reference HXB2 sequence) which is between the NFκB binding sites and the transcription start site (TSS) [41]. Chromatin immunoprecipitation (ChIP) followed by qPCR was used with four primer pairs against the 5′LTR, the first transcribed nucleosome *nuc-1*, further downstream *gag* and the human gene *rpp30* as reference. The data confirm that the ZFP3 binds preferentially to the LTR region of the provirus (Fig 1C). The ChIP generated low DNA yield as expected from the low ZFP3-FLAG abundance and high binding specificity. We also performed ChIP followed by massive parallel sequencing (ChIP-seq) to identify other regions in the human genome where the ZFP3 protein might bind. In replicate experiments ($n$ = 2) we performed MACS2 peak calling (S1 Table). Filtering the data (FDR$<10^{-5}$, in both replicates) identified nine peaks (Fig 1D). A more stringent filtering (FDR$<10^{-10}$ and peak width$<$1kb) identified the HIV-1 provirus as the single binding site of ZFP3, at a peak from 293–470 (9,327–9,556) overlapping the expected binding sequence. This confirmed that the ZFP3 protein uniquely marks the HIV-1 provirus in 1C10 cells, which enables the use of the PLA technique to study the microenvironment of the HIV-1 provirus in single cells.

## Detecting HIV-1 activation by proximity of Tat and ZFP3

Our first aim was to detect early physiological activation of the HIV-1 provirus by investigating the recruitment of Tat to the HIV-1 promoter. We performed PLA with antibodies against Tat and the FLAG-epitope of ZFP3 in the 1C10 J-lat cells (Fig 1E). Nuclei were microscopically identified (500–5,000 nuclei per reaction) and DAPI intensity was used as proxy for the DNA content to estimate the cell cycle stage [45]. In the orange channel for PLA, foci were detected at a range of thresholds for the fluorescent intensity (S1B Fig). Based on control experiments, under the premise that Tat should be at the HIV promoter at higher levels after PMA/i-activation compared to before activation, we determined a threshold for the intensity of the PLA focus that reflected this difference (S1B Fig). This threshold was used throughout the experiments, but slightly adjusted in few samples to account for experimental variability. The cells with a nucleus containing a single focus were considered PLA$^+$. Rare nuclei with multiple foci were not counted.

Upon 16 h of treatment with PMA/i to stimulate the cells and activate the provirus, 6.3 ±1.2% ($n$ = 5) of the cells were PLA positive, at the selected threshold. In DMSO treated samples, where Tat was not expected to be expressed, the background level of Tat-ZFP3 PLA$^+$ cells was 1.9±0.2%. using the same intensity threshold (Fig 1F). To test the sensitivity of the PLA method, we included the parental 5A8 cells. In a parallel experiment, Tat-ZFP3 PLA nuclear foci were detected in 1.5±0.1% of the 5A8 cells lacking FLAG-ZFP3 regardless of cellular activation status.

As the HIV-1 provirus in these J-lat cells was linked to a GFP coding sequence, GFP expression was measured in the same cells. The activated cells had 27±5% of GFP$^+$ cells in both 1C10 and 5A8 cells, with a background of 1.5±0.3% in DMSO treated cells (Fig 1G). The unstimulated Tat-ZFP3 PLA$^+$ cells contained spontaneously activated proviruses as well as technical artefacts. These cells did not express GFP above background levels (S1C Fig). Flow cytometry data of the 1C10 cell lines showed 0.19% of unstimulated cells spontaneously expressing GFP

and 19.3% of stimulated cells expressed GFP (S1C Fig), after gating for viable cells. The discrepancy between GFP$^+$ rates from microscopic imaging and flow cytometry indicated that the microscopy settings were more inclusive and detected cells with lowly expressed GFP. The similarity in response to T cell stimulation in the two cell lines differing in ZFP3 status demonstrated that tethering ZFP3 to the HIV-1 promoter did not interfere with the proviral activation. Overall, the Tat-ZFP PLA$^+$ and the GFP$^+$ cells only partially overlapped, as only 5.5±0.9% of GFP$^+$ cells were PLA$^+$ and 36±3% of the PLA$^+$ cells were also GFP$^+$ (Fig 1H). Shifting the PLA threshold had little effect on the GFP intensity in the PLA+ cells (S1D Fig). Together, this suggests that the two read-outs captured different though overlapping aspects of the HIV-1 activation.

As the nuclear position of the provirus can be determined in our samples, we calculated the distance from the PLA spot to the nuclear periphery. As expected from previous studies [15,46–48], most proviruses with promoter-proximal Tat were close to the nuclear envelope (Fig 1I). After T cell stimulation, the provirus was found even closer to the nuclear periphery (S1E Fig). This is consistent with transcriptionally active proviruses having a dynamic nuclear location [49].

In an alternative model system, we used the previously described K562 cells with latent HIV-1 [50,51]. K562 cells with a latent reporter HIV-1-GFP were transfected with nuclease-dead Cas9 (dCas9) together with a guide RNA targeting dCas9 to the 5′-region of the HIV-1 provirus. Cells were then activated, and PLA was performed with antibodies against dCas9 and Tat as previously. GFP was scored together with PLA signal (S2 Fig). The results demonstrate that using a protein anchor such as a ZFP or dCas9 together with a guide RNA, the PLA method can be used in a more general context to determine protein proximity to any genomic region in single cells.

## Tat is found mainly at the proximity of the HIV-1 promoter in activated cells

Tat under an ectopic promoter has been reported to have affinity for numerous genomic regions in unstimulated Jurkat cells [35,36]. This raised our concern as high levels of non-HIV-1 Tat potentially could increase unspecific background signal in our experiments. Therefore, we mapped Tat in our HIV-1 containing 1C10 J-lat cells with and without T cell stimulation. J-lat cells increase the Tat protein level after cell stimulation [52]. ChIP-qPCR showed that Tat was preferably found at the HIV-1 promoter in stimulated cells (Fig 2A). To map the genome-wide distribution of Tat in the nucleus, we performed ChIP-seq. Tat peaks were detected by the MACS2 algorithm where we compared anti-Tat to input for both activated and resting cells. In this experiment with duplicate samples, we identified 78 Tat peaks in both replicates of the stimulated cells and no peaks in unstimulated cells (S1 Table). The sequences of the peaks were analyzed for consensus motifs, but none were identified. We performed a GO analysis of the Tat-associated regions and found no significant terms enriched. Among the highly significant regions (FDR<10$^{-5}$ in both replicates), six regions were identified whereof the most significant region was just downstream of the HIV-1 TSS (Fig 2B). The other regions included the splice acceptor within the HIV-1 *env* region, as well as four regions of the host genome. The four non-HIV-1 peaks were found in intergenic regions with no obvious link to HIV-1 or T cell activation. After PMA/i exposure, a minor local accumulation of Tat was found on chr 2 at the *MAT2A* locus (S3A Fig). This is the genomic integration site of the HIV-1 provirus in these J-lat cells [43].

Even though we did not clearly identify non-HIV-1 positions associated with Tat binding, we mapped previously identified regions to our dataset. In a dataset from D'Orso and

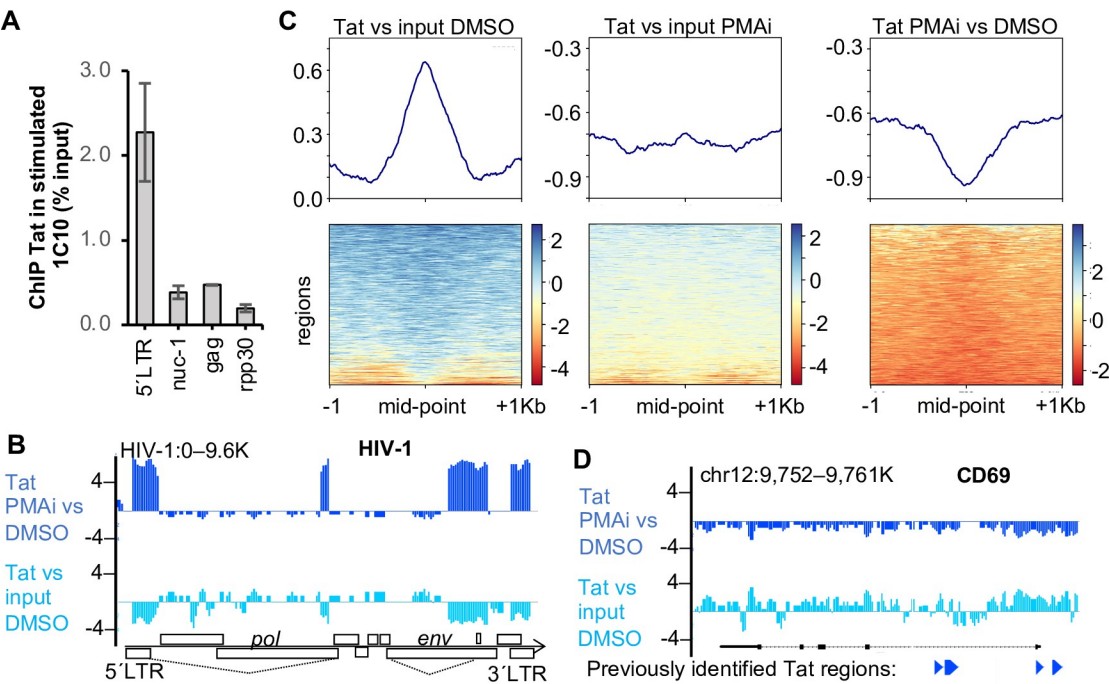

**Fig 2. Tat is predominantly found at the HIV-1 promoter in PMA/i activated cells.** (A) ChIP-qPCR of Tat at three positions of the HIV-1 provirus and control ($n$ = 3, s.e.m.). (B) Genome browser view of Tat ChIP-seq at the HIV-1 locus. (C) Heatmaps of ChIP-seq against Tat in cells unstimulated (DMSO) or stimulated (PMAi) relative to input or stimulated relative to unstimulated ($n$ = 2). (D) Genome browser view of Tat ChIP-seq, the CD69 locus with the exons in black and the Tat-ChIP peaks from Reeder et al (2015) [36] in blue at the bottom.

colleagues, 6,114 peaks were found associated with Tat binding in unstimulated Jurkat cells [36]. The genes associated with these regions were involved in many processes and the most significant processes were regulation of cell aging and cell cycle processes. We indeed observed Tat at these regions in our unstimulated 1C10 cells (Fig 2C). However, in the activated cells, the metagene curve was suppressed, with only background variation remaining. The loss of non-HIV-1 Tat during T cell activation became even more noticeable as the decline of Tat at these regions mirrored the Tat profile in unstimulated cells. As an example of these regions, we specifically assessed the locus encoding CD69 (Fig 2D). The browser view demonstrated noticeable levels of Tat at the CD69 promoter locus in unstimulated cells, as previously noted [36]. However, after cellular stimulation, Tat has been redistributed to other regions. This confirms that in stimulated cells under physiological Tat protein levels, Tat binds predominantly to the HIV-1 provirus.

## PLA and GFP capture different aspects of drug-induced HIV-1 activation

We then sought to determine the effect of latency reversal agents (LRAs) using Tat-ZFP3 PLA. In contrast to other methods that detect transcription irrespective of Tat status, we uniquely detect Tat-dependent transcription. Here we assessed latency reversal 16 h after exposure to seven commonly studied LRAs, alone or in combination and at different concentrations. These drugs were the protein kinase C agonists bryostatin, PEP005 and prostratin, histone deacetylase inhibitors romidepsin and panabinostat, BET bromodomain inhibitor JQ1, and the DNMT inhibitor 5-aza-2′-deoxycytidine (5azadC) (Fig 3A). The experiment was controlled by PMA/i-induced T cell activation. We compared the Tat-ZFP3 PLA results to the

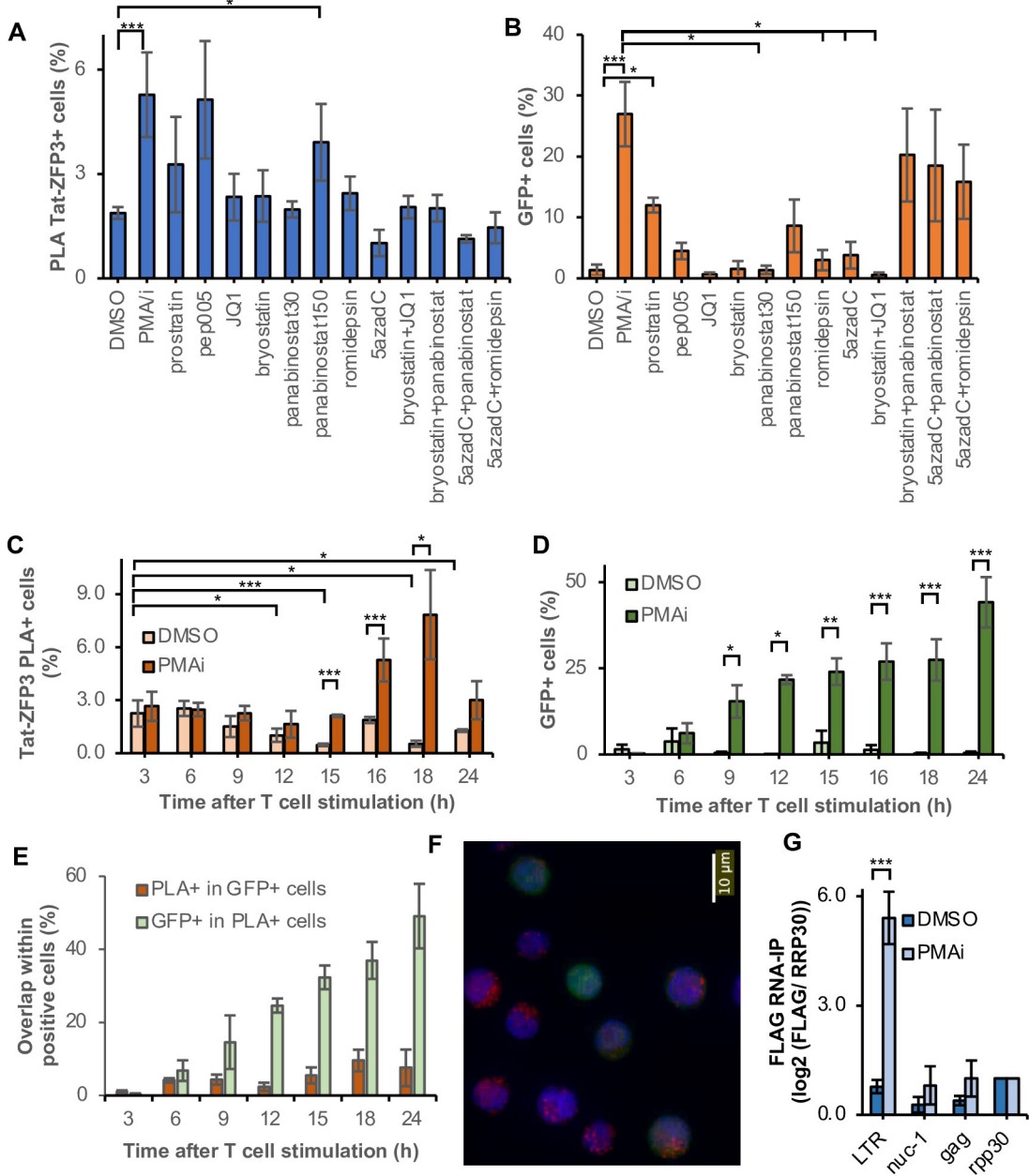

**Fig 3. Tat is recruited to the HIV-1 promoter early after latency reversal and T cell activation but binds to cytoplasmic HIV-1 RNA at late timepoints.** (A) (C) Time-series with Tat-ZFP3 PLA spots (n = 4). (D) Time-series with GFP. (E) overlap between GFP $^+$ and Tat-ZFP3 PLA$^+$ cells. (F) Micrograph of DAPI-stained (blue) 1C10 cells 24 h after PMA/i exposure. ZFP3-Tat PLA spots in red, GFP in green. (G) RNA-immunoprecipitation (RIP) using anti-FLAG. 1C10 cells containing ZFP3-FLAG were treated 24 h with PMA/i or DMSO, qPCR was performed using three primers in HIV-1 and normalized to human *rpp30* (n = 3, error bars represent s.e.m.).

expression of GFP in the same cells (Fig 3B). The GFP measurements based on PLA largely recapitulate our previous results from flow cytometry in the parental 5A8 cell line [21]. T cell stimulation by PMA/i reactivated HIV-1 as detected by both PLA and GFP. However, among the potential LRAs, only the high dose of panabinostat (150 μM) significantly reversed latency detected in both read-outs. Low correlation (0.25) was determined between the GFP and PLA

signals. Least-square linear regression resulted in a goodness-of-fit ($R^2$) of 0.5 (S3B Fig). This suggests that different aspects of latency reversal are captured by the two methods.

## The dynamics of Tat-promoter interactions during cellular activation

To determine the dynamics of Tat recruitment to the HIV-1 promoter during T cell stimulation, we performed a time-series experiment. During 24 h after cellular exposure to PMA/i or DMSO, the appearance of PLA spots representing Tat in the vicinity of the promoter was recorded. As expected, the frequency of cells with a single Tat-ZFP3 PLA spot in the nucleus increased in time in the stimulated cells (Fig 3C). Interestingly, an early background signal could be detected that peaked at 6 h after DMSO addition and then decreased. Probably this observation arose as an effect of cell handling at the beginning of the experiment. At 15 h after addition of PMA/i, the Tat-ZFP3 PLA spots were above the background of DMSO-treated cells. However, after 18 h, the frequency of Tat-ZFP3 PLA$^+$ cells no longer increased. In contrast, the frequency of GFP$^+$ cells continuously increased from 9 h after T cell stimulation (Fig 3D). We also examined the overlap between Tat-ZFP3 PLA$^+$ and GFP$^+$ cells (Fig 3E). Whereas the GFP$^+$ cells did not gain PLA foci in time, the fraction of Tat-ZFP3 PLA$^+$ the expressed GFP gradually increased. This further supports the interpretation that Tat is recruited to the promoter early, and after a delay, viral proteins are produced.

## Apart from HIV-1 promoter DNA, ZFP3 also binds to the HIV-1 RNA sequence

At late time points, notably at 24 h after T cell stimulation, we noted irregularities in the PLA data. This was not observed in the unstimulated cells. Upon close examination, we noted a high fraction of nuclei with multiple PLA signals in some of the samples (Fig 3F). PLA spots appeared in the cytoplasm and even outside of the cell, and therefore were not recorded as Tat-ZFP3 PLA$^+$ cells earlier. We hypothesize that these clusters of PLA signal were caused by ZFP3 binding to the RNA sequence, apart from binding its target sequence in nuclear DNA. Consequently, Tat-ZFP3 PLA signal would also be detectable in the cytoplasm and in viral particles if ZFP3 binds to the LTR RNA sequence. To test this hypothesis, we performed RNA-immunoprecipitation (RIP) using an anti-FLAG antibody. The results show that the pulled down FLAG-ZFP3 is associated with RNA from the LTR region of HIV-1 in activated cells (Fig 3G). The ZFP3 binding to the LTR RNA sequence was confirmed RNA electrophoretic mobility shift assay (EMSA) (S4 Fig). First, a plasmid expressing ZFP3 and a parental plasmid without the ZFP3 sequence were transfected into 293T. In lysates from cells infected with a ZFP3-expression plasmid, a labelled RNA probe was shifted on the gel, implying ZFP3-RNA interaction. The shifted band was specifically competed out by the addition of an unlabeled probe (S4A Fig). This shift was not detected in lysates from cells expressing a non-ZFP3-containing plasmid (S4B Fig). We concluded that the observed PLA signals at late time point originated from ZFP3 interaction with RNA. Specifically, the interaction must occur at the 3′ LTR, as the TSS is downstream of the 5′LTR ZFP3 binding sequence, and thus not present in the HIV-1 RNA. As we are interested in Tat's role in the initiation of transcription, we decided to perform the following Tat-ZFP3-PLA experiments at 16 h, before viral RNA accumulation in the cytoplasm.

## Activated cells accumulate in G1 while Tat is promoter-proximal in S-G2

Another observation we made here was that, compared to the unstimulated cells, fewer stimulated cells were recovered at late time points despite same starting material. The 1C10 cells have a population doubling time of 20 h, resulting in the cell numbers more than double in the

24 h following culture addition of DMSO. However, cells grown in media with PMA/i remained at similar numbers after 24 h. This has previously been described as a consequence of a balance between activation-induced proliferation and AICD [38]. At 24 h, the growth of the 1C10 J-lat cells in PMA/i containing media was 42±5% of the DMSO control. In addition, the viability was 83% in PMA/i treated cells compared to 99% for DMSO treated cells as evaluated by live-dead stain and flow cytometry (S5A Fig). Together, at 24 h after PMA/i-mediated activation, the surviving fraction was derived from 30% of the original cells, thus the cell population at this time point did not represent the original population. We measured apoptosis and necrosis by annexin V and propidium iodine staining using flow cytometry. At 16 h after T cell stimulation, apoptosis was induced by PMA/i in both J-lat 5A8 cells and parental Jurkat cells (Fig 4A). Levels of necrosis remained low. Other agents to stimulate HIV latency reversal

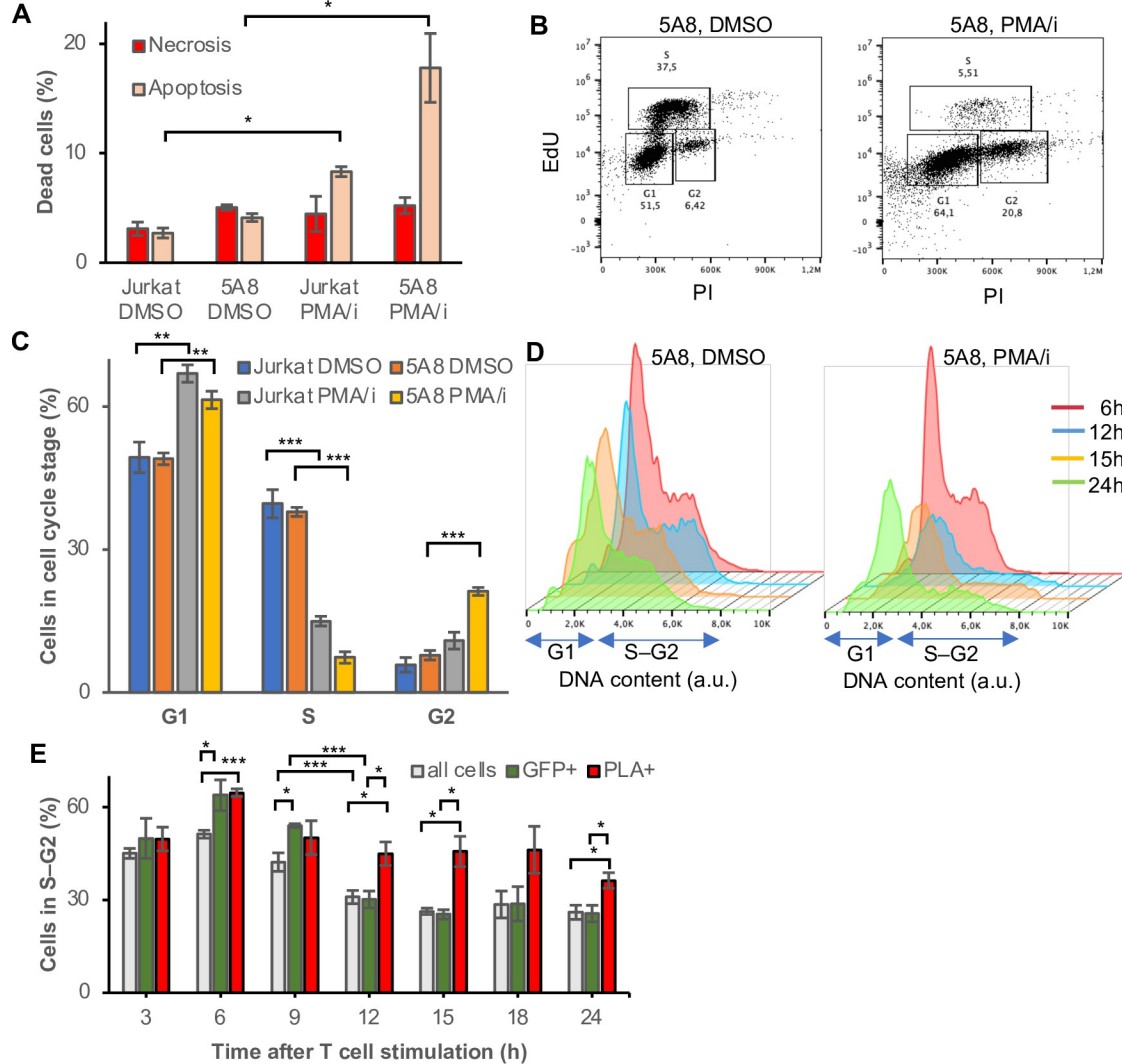

**Fig 4. Tat is transiently bound to HIV-1 promoter in all cell cycle stages.** (A) Apoptosic and necrotic cell populations determined by PI/Annexin V staining and flow cytometry in J-lat 5A8 cells and parental Jurkat cells after 16 h T cell stimulation with PMA/i. (B) Incorporation of thymidine homolog EdU during a 1 h pulse 16 h after PMA/I stimulation. (C) (D) DNA content was calculated from DAPI intensity in unstimulated cells (DMSO) (left) or stimulated cells (PMA/i) (right) from the micrographs at 9–24 h PMA/i exposure. (D) Percentage of cells in S–G2 phase of the cell cycle at different time points after T cell stimulation. All cells in grey, GFP+ cells in green, and Tat-ZFP3 PLA+ cells in red, $n = 5$, error bars represent s.e.m., Student t-test p-values * $p < 0.05$, ** $p < 0.01$, *** $p < 0.005$.

were not toxic and only in HIV-1 containing 5A8 cells did CD3/CD28 activation or prostratin induce apoptosis (S5B Fig). Given the viral cytopathic effects of HIV-1 proteins, we expected the surviving fraction to be enriched in cells that have expressed no, low or transient HIV-1 levels.

To further investigate the connection between T cell stimulation and Tat-dependent HIV-1 latency reversal, we analyzed the cell cycle profiles. Firstly, we quantitated cell cycle progression by incorporating thymidine homolog EdU (5-ethynyl-2′-deoxyuridine) in replicating DNA during 1 h (Fig 4B). Compared to unstimulated cells, 16 h of PMA/i stimulation led to a cell cycle arrest in both Jurkat and 5A8 cells. Both cell types accumulated in G1, but 5A8 cells also accumulated in G2 (Fig 4C). To link cell cycle to the Tat recruitment to the HIV-1 promoter, we analyzed the cell cycle profile of the cells on the PLA slides. Microscopically, the DNA content was used as a proxy for cell cycle stage [45]. In DMSO, the cell cycle profile remains similar throughout the 24 h time course, whereas after 12 h of PMA/i stimulation, cell proliferation was reduced as indicated by a decrease in S–G2 (Fig 4D), coupled with an increase of cells in G1 and also an increase in dying cells as indicated cells with sub-G1 DNA content (S1C and S1D Fig). The profile of Tat-ZFP3 PLA+ and GFP+ cells appeared different (Fig 4E). Tat at the HIV-1 promoter was consistently overrepresented in the S–G2 fraction, as was spontaneous or early PMA/i-induced GFP activation. This was expected from HIV-1 latency reversal being linked to metabolically active cells [51]. However, at later time points, cells in general together with GFP+ cells accumulated in the G1 phase (S5C Fig). With the G1/S arrest activated through the AICD process in the cells, cells with continuous activation of the HIV-1 would be expected to enrich in the G1 fraction, as we observed in the GFP+ cells. However, the PLA+ cells displayed a different pattern. At the 15 h time point, before the confounding factor of cytoplasmic spots occurred, 46±5% of the Tat-ZFP3 PLA+ cells were found in the S–G2 phases, compared to 26±1% of all cells found in late S-G2 (Fig 4E).

## The chromatin composition of the HIV-1 promoter reflects latency reversal potential

Apart from detecting of Tat at the HIV-1 promoter, we can also use the ZFP3 PLA assay to characterize the microenvironment of the HIV-1 provirus in other aspects, e.g., to determine the suggested heterogeneity of chromatin marks in the reservoir of latently infected cells [5,21,53]. Here we queried the active marks H3K4me1, H3K4me3 and H3K27ac as well as heterochromatin marks H3K9me3 and H3K27me3 in addition to total H3 (Fig 5A). For the first time, we present direct single cell data of chromatin on the reversal of HIV-1 latency. H3 is expected to be detected in all cells. However, we only detected a H3-ZFP3 PLA signal in 6.0 ±1.8% of DMSO treated cells and in 11±2% of cells after PMA/i exposure. This increase most likely reflects a more accessible chromatin after activation. This low recovery prevents comparisons between antibodies, and rather suggests that comparisons should be made between unstimulated and stimulated cells. Strikingly, the activating chromatin marks, H3K4me3, H3K4me1 and H3K27ac seemed not affected by stimulation whereas the heterochromatin marks, notably H3K9me3 was detected in more cells after T cell stimulation. H3K9me3 was the only mark that was significantly (p<0.005) different in stimulated compared to unstimulated cells. Worth noting, these data were generated using the surviving fraction of cells that were still intact after 16 h of activation. As we determined previously, T cell stimulation decreases viability and the toxic viral proteins after HIV latency reversal further reduce viability of virus-producing cells. The fraction of cells present on the coverslips were likely not to have produced large quantities of virus yet. These results are consistent with H3K9me3 at the provirus preventing HIV-1 reactivation and the toxic effects of viral expression.

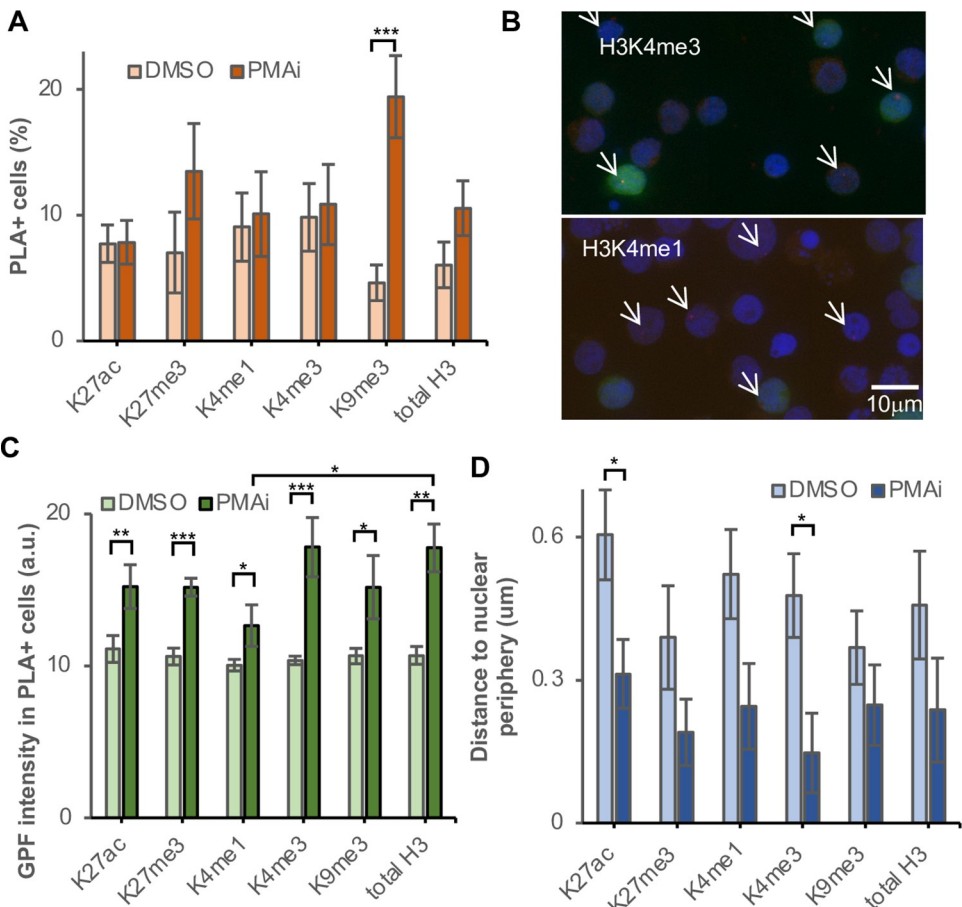

**Fig 5. Non-permissive chromatin marks increase in the surviving population after T cell stimulation.** (A) H3 modifications found at the HIV-1 promoter at 16 h post T cell stimulation. (B) GFP expression in cells with HIV-1 associated with different marks. (C) Quantification of GFP. (D) Distance to nuclear periphery of the HIV-1 locus. P-values * p<0.05, ** p<0.01, *** p<0.005.

The H3K9me3 mark is a repressive mark but repression depends on the binding of heterochromatin protein 1 (HP1). Different isoforms of HP1 exist. Whereas HP1α and HP1β prevent promoter activity, HP1γ prevents transcription initiation but occupies regions downstream of the promoter in expressed genes, including HIV-1. To test the mechanism whereby the provirus in H3K9me3 chromatin was silenced, we probed for the HP1α and HP1γ isoforms. None of these could be detected in our cell model, as PLA signals were consistently below 0.6% of cells in both unstimulated and stimulated cells.

Given the nature of the experimental system, we had simultaneous access to the GFP expression and PLA signals in the same cells (Fig 5B). To test the reactivation potential associated with the histone modification, we compared the induction of GFP in the ZFP3 PLA+ cells after T cell stimulation (Fig 5C). Cells with proviruses associated with all the tested histone modifications had significantly (p<0.05) higher GFP levels after activation. H3K4me3 had the highest relative GFP induction, followed by H3. The weakest evidence for GFP induction was detected with H3K9me3 and the enhancer mark H3K4me1. The GFP level in activated cells was even significantly (p<0.05) lower in H3K4me1-containing proviruses compared to total H3.

We then determined the distances of the HIV-1 proviruses to the nuclear periphery (Fig 5D). Proximity to the nuclear pore facilitates the export of the unspliced RNA required for virus production. The marks of active promoters, H3K4me3 and H3K27ac, were both significantly ($p<0.05$) closer to the nuclear periphery after T cell stimulation.

## Enhancer H3K27ac is required for Tat-mediated HIV-1 activation

We have previously found that the marks of active enhancers–H3K4me1 in combination with H3K27ac–have a repressive function on HIV-1 [21]. Notably, pre-exposure to a drug GNE049 acted as a weak LRA but resulted in drastically reduced levels of GFP after PMA/i stimulation of J-lat cells. Employing the ZFP3 PLA tool, we wanted to expand on our previous observation that enhancer-like chromatin modifies the activation status of the HIV-1 provirus. We treated the cells with two enhancer-modifying drugs: a BRD4 inhibitor JQ1, and GNE049 that at low doses act as a specific small molecule inhibitor of CBP/P300-mediated H3K27ac at enhancers, but that at higher doses also inhibits BRD4 [54,55]. GNE049 and JQ1 exposure led to a global decrease in H3K27ac (Fig 6A). Specifically at the HIV-1 provirus, GNE049 reduced the H3K27ac levels but JQ1 had no effect (Fig 6B). HIV-1 was still inducable by PMA/i after pre-exposure to GNE049 and JQ1 as the GFP levels were induced as in the control T cell

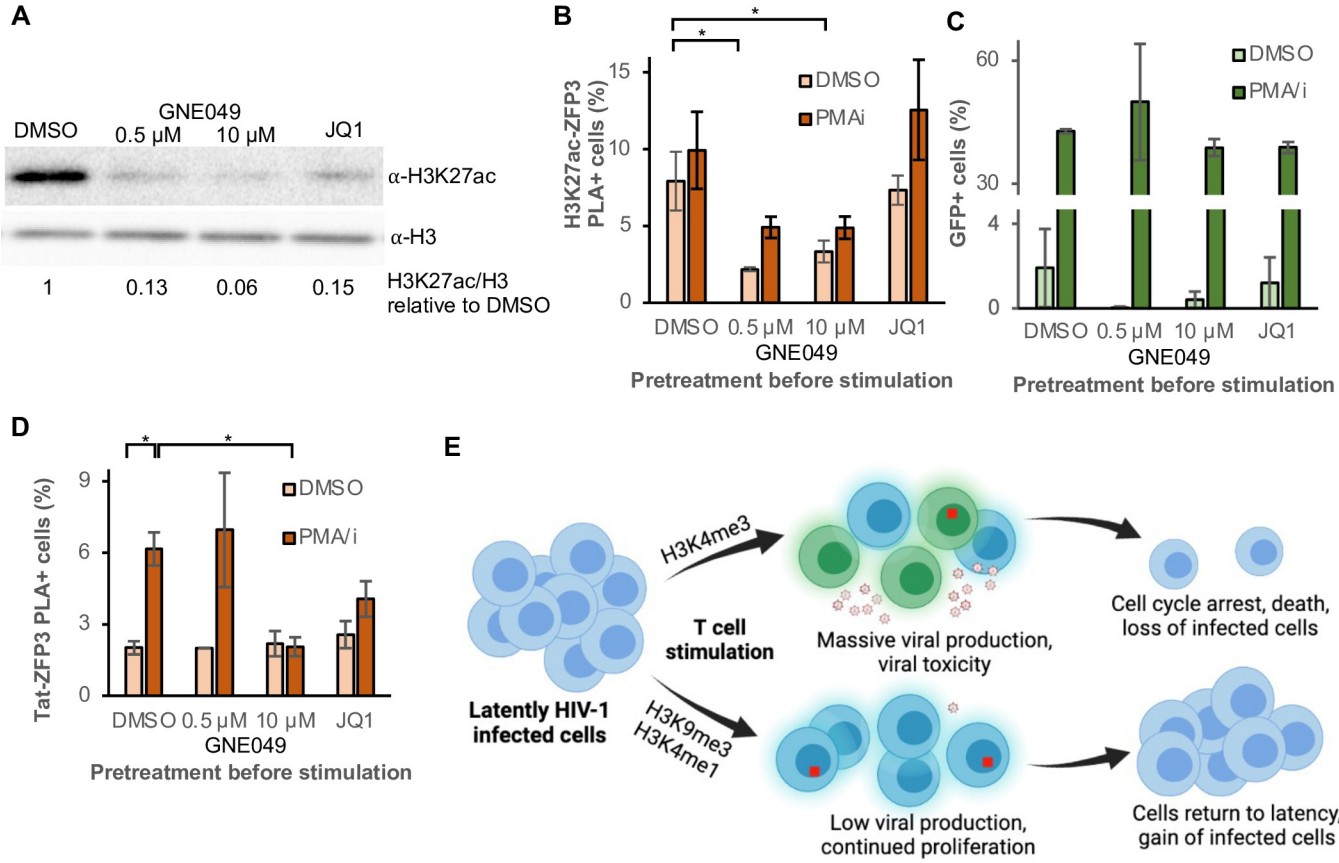

**Fig 6. GNE049 reduces H3K27ac and prevents Tat at the promoter without affecting HIV-1 activation.** (A) H3K27ac-ZFP3 PLA in cells pretreated with GNE049 (a CBP/P300 inhibitor of enhancer H3K27ac) or DMSO for 3 h followed by 16 h with DMSO or PMA/i. (B) Tat-ZFP3 PLA in the same cells as in (A). (C) Percentage of total GFP+ cells treated as in (A). (D) Model of different reservoir compartments after T cell stimulation: upper) high viral production, both dependent and independent of Tat, toxicity leads to cell cycle arrest and cell death; lower) low Tat-mediated viral production in cells with proviral H3K9me3 or H3K4me1, where cells proliferate and return to latency.

stimulation (Fig 6C). However, after high dose exposure to GNE049 blocking both CBP/P300 and BRD4, Tat was no longer recruited to the promoter after activation (Fig 6D). At the low dose of GNE049, no effect was detected on HIV-1 latency reversal even though the H3K27ac levels were low. This indicates that Tat recruitment requires both H3K27ac and BRD4 at the HIV-1 promoter. The results show that 5A8 J-lat cells signal HIV-1 latency reversal through GFP despite the lack of Tat at the promoter, but an enhancer structure of the provirus is required for a Tat-dependent latency reversal of HIV-1.

## Discussion

Reversing proviral latency in the HIV-1 reservoir is highly complex. Even in a homogeneous population of cells, intercellular differences result in seemingly stochastic proviral reactivation. The processes leading to latency reversal are highly interconnected because of several positive feedback loops, *e.g.*, Tat regulating its own transcript and HIV-1 inducing cell death in bystander cells that in turn activates HIV-1. To disentangle the different activities, we present a single cell method that uniquely identifies a transient pulse of Tat in an early but essential step during HIV-1 latency reversal. Without Tat enforcing the HIV-1 transcription, cells do not shed viral particles. The Tat-ZFP3 PLA technique can selectively identify productive, Tat-dependent HIV-1 activation among spurious and non-physiological Tat-independent activation, that might be a consequence confounding factors such as cell stress. In addition to providing an early readout for HIV-1 latency reversal, the proviral chromatin microenvironment can be interrogated in individual cells, together with recording accumulation of GFP representing total HIV-1 production. The method is well suited to untangle mechanisms underlying latency reversal. The observed discrepancy between GFP and Tat-ZFP3 can arise from two mechanisms: 1) Tat transiently associating with the TAR region during early latency reversal followed by GFP accumulation, or 2) GFP being expressed through Tat-independent mechanisms.

Pre-treatment of the cells with GNE049 or JQ1, is likely to result in Tat-independent HIV-1 latency reversal, as Tat-ZFP3 PLA[+] foci could not be detected above background in these cells. This result indicates that the previously identified enhancer structure at the HIV-1 promoter is required for Tat recruitment. The two molecules GNE049 and JQ1 inhibit either enhancer-specific CBP/P300-mediated H3K27 acetylation or BRD4 binding to acetylated histone H3. BRD4 inhibitor JQ1 has been studied previously as an LRA, both alone and in combination with T cell activation [56,57], and we have reported an effect on HIV-1 reactivation of GNE049 earlier [21]. Given the role of both agents to interfere with protein acetylation, the observed effect possibly stems from reduced Tat acetylation. However, diminished Tat acetylation is expected to result in added Tat at the HIV-1 promoter [31,32] as opposed to the observed Tat absence. Consistent with our results, another study revealed that–in contrast to PKC agonists or HDAC inhibitors that reverse latency of proviruses in gene-like chromatin structures–BET inhibitors act on a separate fraction of proviruses, possibly proviruses within enhancer-like structures [11]. By blocking Tat-mediated transcription of enhancer chromatin, the fraction of cells that produce long viral transcripts seem not affected, although the transcription efficiency seems reduced [21]. However, given the well-described role for Tat in HIV-1 transcription, these transcripts are unlikely to produce viruses to spread the infection [29,33,34].

The HIV-1 reservoir persists through proliferation [58]. Our observation that the cells with Tat at the promoter were not affected by the G1/S arrest (S5C Fig) suggest that Tat is transiently recruited to the HIV-1 promoter. From this analysis we conclude that, after T cell stimulation, GFP detects the dying or arrested cells, whereas Tat-ZFP3 PLA detects a fraction of

proliferating cells. Proliferating cells will preserve and expand the pool of HIV-1 infected cells post-activation. This proliferation is driven both by antigens and homeostasis [26]. While the decay of the reservoir is slow, the proviral chromatin is dynamic. Even in the absence of activating signals, the cellular fraction with heterochromatin and enhancer chromatin structures at the provirus expands [21]. We have shown that even in a rather homogeneous cell population, a multitude of different mutually exclusive chromatin structures are present at the HIV-1 integration site. None of the H3 modifications tested here completely prevented HIV-1 reactivation and even proviruses encapsulated in either of the heterochromatin marks H3K27me3 or H3K9me3 produced mRNA after T cell stimulation. H3K27me3 is associated with bistable chromatin and recently it was shown that depletion of PRC1 lead to rapid derepression of transcription with maintained H3K27me3 [59]. Even though we detect H3K9me3 at the provirus, we did not detect HP1α or HP1γ that would condense H3K9me3 chromatin in these cells, possibly explaining the preserved HIV-1 activation potential. However, the weakest induction of GFP came from proviruses found in H3K27ac or H3K9me3 heterochromatin consistent with these proviruses causing the least cytotoxicity.

Based on the data presented here, we propose a model (Fig 6D) of how the proviral chromatin composition affect the latently HIV-1 infected T cells after stimulation. T cell stimulation leads to initial cellular expansion and availability of transcription factors including NFκB. Proviruses present in a chromatin confirmation resembling an active or poised host gene and thus associated with H3K4me3 at the promoter will be transcribed, resulting in viral mRNA production and protein production. Viral cytotoxicity will also induce cell cycle arrest and AICD. In the presence of ART, the new viruses will not be able to infect other cells and this fraction of the latent reservoir will diminish. On the other hand, cells with a different chromatin composition at the provirus, *e.g.*, H3K9me3 or H3K4me1 will activate the virus to a lesser extent upon T cell stimulus. Thereby these cells will induce apoptosis to a lesser extent. On the contrary, they have the potential to proliferate resulting in an expanded fraction of the reservoir. In this balance, the number of HIV-1 infected cells remain similar, but the HIV-1 reservoir gradually enters a deeper latency. In time, after spontaneous HIV-1 latency reversal and occasional activation of the CD4 cells, this would shape the HIV-1 reservoir, and the initially rare H3K9me3 proviruses becomes more prominent. Recently, a progressive selection of transcriptionally silent proviruses was shown during 12 years of ART [27]. A progressive increase of silenced proviruses combined with a reduction of easily activatable proviruses would explain the appearance of post-treatment controllers after long-term ART, despite the number of HIV-1 infected cells remining similar [60]. Even though the proviruses in inaccessible H3K9me3 expand, this may not prevent a functional HIV-1 cure [8,27]. However, the more worrisome sub-compartment of the reservoir consists of cells with proviruses embedded in enhancer-like structures. These proviruses are in a stable silent state that remains open and accessible. Short transcripts are continuously produced, yet they are invisible to the immune system. As a transient pulse of Tat allows proliferation of the HIV-1 infected cells, followed by Tat removal to restores HIV-1 latency [33], these cells are likely to be responsible for rebound during ART interruptions [61,62]. Specifically eliminating this subpopulation of cells containing latent but reactivatable HIV-1 will reduce the fraction of the reservoir responsible for rebound viremia. We show here that, small molecules such as GNE049 or JQ1 can remove the enhancer functionality of the provirus, so that when the T cell is activated, HIV-1 latency reversal within the H3K4me1 compartment of the reservoir is Tat-independent, and thus unable to contribute to new infections but still producing cytotoxic viral proteins and potentially exposing the cell to the immune system for elimination.

The method presented here has two main limitations. Firstly, by relying on an ectopic protein–an engineered zinc-finger protein or a nuclease-dead Cas9 linked with a guide RNA to

label the genomic locus in live cells–the method is currently limited to cells that can be transfected. In contrast to the proliferating cells used here, the latent reservoir *in vivo* exists in hard-to-transfect resting CD4 cells. Secondly, the background of the Tat-ZFP3 PLA signal (1.5 ±0.1% of cells lacking ZFP3) prevents reliable detection of rare proviruses in the latent reservoir of ART-treated individuals, where HIV-1 is found in less than 1/10,000 cells [63]. Consequentially, we have used cell lines to model HIV-1 latency reversal. In our optimized protocol, much of the background PLA signal appears to be connected to the Tat antibody, as the antibodies against the HP1 proteins yielded background signal one order of magnitude lower. The Tat protein is highly unstructured and antibodies against Tat also recognize non-Tat targets as noted by us (Fig 2) and others [36,64]. We set an arbitrary but conservative threshold based on the results from negative controls from 5A8 cells (that do not express FLAG) and unactivated 1C10 cells (that do not express Tat). As a consequence of using a conservative threshold, Tat-dependent transcription might be more prominent than we observe. However, relaxing the threshold did not lead to larger overlap with GFP$^+$ cells (S1D Fig), confirming that Tat-independent transcription is commonly observed in these model cells. As Tat-independent transcription does not lead to infectious particles *in vivo*, this could explain the discrepancy between studies of LRA-induced HIV-1 activation model cells and the effect of LRAs in clinical studies [2–4].

To conclude, we present a technique for investigating the chromatin microenvironment at a predetermined locus in single cells. We apply this technique to study HIV-1 proviral chromatin and mechanisms of latency reversal. This allows us to dissect Tat-dependent HIV-1 activation without confounding effects such as cytotoxicity from viral proteins and reveal a role for enhancer-like proviral chromatin that may be exploited therapeutically.

## Material and method

### Plasmid construction

The ZFP3 sequence recognizing CGAGCCCTCAGATGC was synthesized by Genescript (NJ, USA). It was cloned into the plasmid pHR-SFFV-KRAB-dCas9-P2A-mCherry by substitution of KRAB-dCas9 cassette using Gibson assembly (New England Biolabs, Cat#E2611). Later the mCherry was substituted with BFP [50]. Primers for Gibson assembly are found in S2 Table. pHR-SFFV-KRAB-dCas9-P2A-mCherry (Addgene plasmid #60954) and pU6-sgRNA-EF1Alpha-puro-T2A-BFP (Addgene plasmid #60955) were gifts from Jonathan Weissman. Guide RNAs for dCas9 were cloned into the pU6-sgRNA-EF1Alpha-puro-T2A-BFP plasmid.

### Cell culture

J-lat 5A8 and 1C10 cells were cultured in cytokine-free media (RPMI 1640 medium (Hyclone, Cat# SH30096_01), 10% FBS (Life Technologies, Cat# 10270–106), 1% Glutamax (Life Technologies, Cat# 35050), 1% Penicillin-streptomycin (Life Technologies, Cat# 15140–122)).

### Virus production

pHR-SFFV-ZF3-P2A-BFP, psPAX2 (Addgene, Cat#12260) and pMD2.G (VSV-g) (Addgene, Cat#12259) plasmids were purified with Plasmid Plus Maxi Kit (Qiagen, Cat# 12963). psPAX2 and pMD2.G were gifts from Didier Trono. 293T cells (ATCC, CRL-3216; CVCL_0063) grown in DMEM media (Hyclone, Cat# SH30022_01) were transfected with Lipofectamine LTX with PLUS reagent (ThermoFisher, Cat# 15338100), and after 48 h supernatants were harvested. We determined virus titers (p24) by Lenti-X GoStix Plus (Takara Bio Cat# 631280). 5A8 cells were transduced with 50 ng p24/10$^6$ cells. After 3 days, cells plated in 96-well plates

for monoclonal cultures. After 14 days, several clones were tested for BFP expression and GFP expression with and without PMA/i exposure.

## Flow cytometry

Cells were stained with LIVE/DEAD Fixable Violet Dead Cell Stain (Thermo Scientific, Cat# L34955) and fixed in 2% formaldehyde for 30 min. Flow analysis was performed on a Cyto-FLEX S (Beckman Coulter). Individual flow droplets were gated for lymphocytes, viability, and singlets. Data were analyzed by Flowjo 10.1 (Tree Star).

## Apoptosis and cell cycle analysis

For apoptosis staining, cells were stimulated with the respective agent or incubated with DMSO for 16 h. After the incubation period, cells were washed with cold PBS containing 1% BSA and resuspended in Annexin binding buffer (10 mM Hepes adjusted to pH 7.4, 140 mM NaCl and 2.5 mM CaCl). Propidium iodide (50 ng/sample, Biolegend, Cat# 421301) and Annexin V (5 μl, Pacific Blue conjugated, Biolegend, Cat# 640917) were added and the cells were incubated for 15 min at 4°C in the dark before analysis on a Cytoflex S (Beckman Coulter). Staining for flow cytometry was performed on ice in PBS containing 1% BSA. For the exclusion of dead cells, the Fixable Violet Dead Stain Kit (Thermo Scientific, Cat# L34955) was used. Cell cycle analysis was performed with the Click-iT Plus EdU Alexa Fluor 647 Assay Kit (Thermo Scientific, Cat# C10635) according to the manufacturer's protocol.

## Proximity ligation assay (PLA)

Cells were adhered to coated coverslips and a slightly modified PLA protocol was followed. The low abundance of both Tat and FLAG in the cells prompted us to reduce the background signal of the method. Most notably the concentration of enzymes and antibodies were lowered. In short, cells were washed with PBS and allowed to settle onto poly-l-lysine coated coverglasses (Corning Biocoat Cat#354085), marked with a hydropbobic barrier using A-PAP pen (Histolab, Cat# 08046N). PLA was performed according to the manufacturer's protocol (Sigma-Aldrich, cat#Duo92007) with a few modifications: PLA plus and minus probes were diluted 1:20, amplification buffer (5×) was used at 10×. All washes were performed in PBS. Antibodies (1:1,000) were used against Tat (Abcam Cat#43014), FLAG M2 (Sigma-Aldrich Cat# F1804 lot SLCF4933), H3 (Abcam, Cat# ab1791), H3K4me1 (Abcam, Cat# ab8895), H3K4me3 (Diagenode, Cat# C15410030), H3K9me3 (Abcam, Cat# ab8898), H3K27me3 (Diagenode, Cat#C15410069), H3K27ac (Abcam, Cat# ab4729). Before DAPI staining (Life technologies, Cat# 62248) and mounting with ProLong Gold Antifade (Thermo Scientific, Cat# P10144), FITC-conjugated anti-GFP (Abcam, Cat#ab6662) was applied (1:500) for 1 h at ambient temperature protected from light. Slides were sealed with nail polish and stored at 4°C overnight before imaging. Slides were imaged using a Pannoramic Midi II slide scanner (3DHistech) and images were exported using the CaseViewer application. Images were analyzed with ImageJ (version 2.0.0-rc-69/1.52) and macros developed in house. Each RGB image was separated into separate channels. Based on the blue (DAPI) channel, nuclei were identified. In the red channel, spots ("maxima") were detected with thresholds 6–48.

## Chemicals to induce proviral activation

Cells were exposed to latency-reversal agents for 16 h. Drugs and chemicals used were phorbol 12-myristate 13-acetate PMA (Sigma-Aldrich, Cat# 79346) final concentration 50 ng/ml, ionomycin (Sigma-Aldrich, Cat# I0634; Lot#106M4015V) final concentration 1 μM, Ingenol-

3-angelate PEP005 (Sigma-Aldrich, Cat# SML1318) final concentration 12 nM, panobinostat (Cayman Chemicals, Cat# CAYM13280) final concentration 30 nM or 150 nM, JQ1 (Cayman Chemicals, Cat#CAYM11187) final concentration 100 nM, bryostatin (Biovision, Cat# BIOV2513) final concentration 10 nM, prostratin (Sigma-Aldrich, Cat#P0077) final concentration 6 μM, GNE049 (MedChemExpress, Cat# HY-108435) final concentration 0.5 μM or 10 μM.

## Immunoblotting

Cell pellets were lysed in lysis buffer (150 mM NaCl, 50 mM Tris, 1% Triton-X100, 1 mM orthovanadate). Protein concentrations were calculated using Bradford assay (Bio-Rad Cat#5000006). Samples were diluted to 1× Laemmli buffer (Bio-Rad, Cat# 1610747) with 10 mM DTT. Samples were heated for 10 min at 70˚C and then loaded on a 12% Mini-protean TGX gel (Bio-Rad Cat# 4561044). The gels were run for approximately 1 h at 100 V. Proteins were transferred onto a PVDF membrane (Bio-Rad, Cat# 1704156) and blocked in 5% skimmed milk in PBS-T for 1 h at room temperature. Afterwards, the membrane was incubated at 4˚C overnight with primary antibodies (1:1000) in 1% milk in PBS-T. We used primary antibodies as above. After washing and addition of secondary HRP-conjugated antibodies, membranes were developed using Pierce ECL Plus Western Blotting Substrate (Thermo Scientific Cat# 32132). Membranes were stripped (Restore plus Western blot stripping buffer Thermo Scientific Cat# 46430) and re-probed using a second set of control primary antibodies. Blots were run and visualized using a Chemidoc Touch V3 Western workflow (Bio-Rad, Cat# 1708381).

## Chromatin immunoprecipitation

ChIP-qPCR was performed using the iDeal ChIP-qPCR protocol (Diagenode, Cat# C01010180). Each ChIP reaction was performed on $2 \times 10^6$ cells. Cells were fixed with 1% formaldehyde for 10 min in room temperature. Sonication was performed at 30 s in eight cycles (Bioruptor Pico, Diagenode, Cat# B01060010). ChIP was performed using anti-FLAG antibody M2 (Sigma-Aldrich Cat# F1804 lot SLCF4933), or anti-Tat (Abcam Cat#43014). ChIP eluates were purified with Wizard SV Gel and PCR clean-up system (Promega, Cat# A9282). Primer sequences are shown in S2 Table. PCR reactions were performed with Power-up Sybr green master mix (2×) (ThermoFisher, Cat#A25742) using 40 cycles on an Applied Biosystems 7500 Fast Real-Time PCR System (ThermoFisher).

## Massive parallel sequencing

DNA samples were quantified with Qubit dsDNA HS Assay kit (ThermoFisher, Cat# Q32851) and libraries were prepared using NEBNext Ultra II DNA library kit. Libraries were sequenced on an Illumina Nextseq 550 (75 cycles, single-end sequencing) at the BEA facility (Huddinge, Sweden), according to the manufacturer's instructions. Raw data from the sequencing reactions (fastq files) were aligned to the hg38 genome assembly with Bowtie2 (version 2.0.6), set to the default parameters. Resulting sam files were converted to bam files using Samtools version 1.4. Bam files were imported into Galaxy (version 21.05, usegalaxy.org) or SeqMonk version 1.47.2. Sequence reads were quality controlled to have MAPQ>20 (Tat).

## Native RNA-IP

$2 \times 10^7$ cells were activated for 24 h with PMA/i or DMSO. Cells were washed in cold PBS, and KCl, EDTA and RNase Out (Thermo Scientific, Cat# 10777019) were added. The cell pellet

was lysed (50 mM Tris pH 8,1% TritonX-100,150 mM NaCl, 1 mM DTT, 2 mM orthovanadate, protease inhibitor cocktail). The cells were rotated in 4˚C for 30 min and centrifugated in 4˚C 13,000 rpm for 20 min. Magnetic beads were washed in TBS-T (20 mM Tris pH 7,4, 150 mM NaCl, 0.05% Tween20), then in buffer NT2 (50 mM Tris-HCl pH7,4, 150 mM NaCl, 1 mM $MgCl_2$, 0.05% TritonX-100, protease inhibitor cocktail). For each reaction, 2 μl antibody and 25 μl washed Pierce magnetic protein A/G beads were rotated 1 h in RT, before adding buffer (NT2. 20 mM EDTA, 1 mM DTT, 200 U/ml RNase Out, PIC) and cell lysate. Reactions were rotated overnight at 4˚C. The beads were washed six times. Trizol (Thermo Scientific, Cat# 15596026) was added, 5 min at ambient temperature, followed by addition of chloroform, 5 min. The reactions were centrifuged, and the upper aqueous phase was transferred to a new tube. RNA was purified (RNA clean and concentrator, Zymo research, Cat# R1017). RNA, random hexamers 50 nM, dNTP, were incubated 5 min at 65˚C, then on ice 1 min. cDNA was generated using Superscript III (ThermoFisher Scientific, Cat#18080093). cDNA was diluted 1:4 and used for qPCR with Powerup Sybr green master mix (2×) (ThermoFisher, Cat#A25742) using 40 cycles on an Applied Biosystems 7500 Fast Real-Time PCR System (ThermoFisher). Primer sequences are found in S2 Table.

## RNA EMSA

293T cells were transfected with the ZFP3 plasmid (this study) or the parental plasmid. After 48 h, cells were lysed. The protocol from LightShift Chemiluminescent RNA EMSA Kit (Thermo Scientific, Cat# 20158) was followed. The sequence of the biotinylated probe is found in S2 Table. Binding reactions were performed in a total volume of 20 μl. Reaction mixtures were loaded on 5% Mini-Protean TBE native polyacrylamide gels (Bio-Rad, Cat# 4565015). Nucleic acids were transferred onto Amersham Hybond-N+ nylon membrane (Cytiva, Caat# RPN203B) by semi-dry transfer (0.8A, 25V, 15 min) using Trans-blot turbo transfer system (Bio-Rad). Crosslinking (120 J/cm2) was performed with a Stratalinker 2400. RNA was visualized on a Chemidoc Touch (Bio-Rad).

## Supporting information

**S1 Table. MACS2 peaks of ChIP-seq using antibodies against FLAG and Tat after treatment with DMSO or PMA/i for 24 h.**
(XLSX)

**S2 Table. Primers used for Gibson assembly, ChIP, RIP, EMSA and dCas9 sgRNA.**
(XLSX)

**S1 Fig. Detecting HIV-1 using proximity ligation assay (PLA).** (A) Flow cytometry of 5A8 and 1C10 cells, showing BFP (V450-A) against SSC (SSC-A). (B) Boxplot with the GFP levels relative to intensity of the PLA spot. (C) GFP intensity in Tat-ZFP3 PLA[+] cells after 16 h treatment with DMSO or PMA/i. Dotted line shows the background cellular GFP intensity. (D) Flow cytometry of 1C10 cells showing GFP (B525-A) against BFP (V450-A) in unstimulated (DMSO) or stimulated (PMA/i) cells after 24 h (E) Distance between the PLA spot and the nuclear periphery, $n = 4$ (F) Genome browser view of *the MAT2A* locus, harboring the HIV-1 provirus.
(TIF)

**S2 Fig. Tat-dCas9 PLA in K562 cells with latent HIV-1.** (A) Micrograph of PLA using anti-Tat and anti-HA in a K562 cell line with dCas9-HA and latent HIV-GFP. Cells were also transfected with a plasmid to express a sgRNA targeting the 5´ region of the HIV provirus. (B) Quantification of Tat-dCas9 PLA[+] cells and GFP[+] cells in time after T cell stimulation by

antibodies against CD3 and CD28.
(TIF)

**S3 Fig. PLA and GFP capture slightly different aspects of HIV-1 activation following LRA.**
(A–B) Response to latency reversal agents (LRAs) in J-lat 1C10 detected by PLA Tat-ZFP3 (A)
and GFP (B). (C) Correlation between GFP and PLA spot. $n = 5$, error bars show s.e.m..
(TIF)

**S4 Fig. ZFP3 binds specifically to LTR RNA.** RNA EMSA using a biotinylated HIV LTR
RNA probe. Red arrows point to the specific band that appears gel shifted by ZFP3. (A) Gel
with no lysate (lane 1) and increasing lysate concentration (lanes 2–4, 5–7, 8–10), and includ-
ing no competitor (-, lanes 2–4), low (+, lanes 5–7) and high (++, lanes 8–10) concentration of
non-biotinylated competitor HIV LTR RNA. (B) Gel with no lysate (lane 1), cell lysates from
293T cells transfected with ZFP3+ (lanes 2–5) and parental ZFP3- (lanes 6–8) plasmid.
(TIF)

**S5 Fig. Cells accumulate in G1–following PMA/i-mediated stimulation.** (A) Percentage of
live 5A8 cells after T cell stimulation recorded by live/dead cell stain using flow cytometry
($n = 3$). (B) Apoptosic and necrotic cell populations determined by PI/Annexin V staining and
flow cytometry in J-lat 5A8 cells and parental Jurkat cells after 16 h T cell stimulation with
CD3-CD28 antibodies or prostratin. * $p<0.05$ compared to DMSO-treated control. (C) Per-
centage of 1C10 cells in G1 phase of the cell cycle. All cells in white, Tat-ZFP3 PLA[+] cells in
red, GFP[+] cells in green, $n = 5$ error bars represent s.e.m., Student t-test p-values * $p<0.05$.
(TIF)

## Acknowledgments

The authors would like to thank Sara Svensson Akusjärvi for initial work with microscopy. We
would like to acknowledge the core facilities MedH Core Flow Cytometry facility (Karolinska
Institutet) for providing cell analysis services, and BEA, Bioinformatics and Expression Analy-
sis (Karolinska Institutet) for providing sequencing services.

## Author Contributions

**Conceptualization:** J. Peter Svensson.

**Data curation:** J. Peter Svensson.

**Formal analysis:** J. Peter Svensson.

**Funding acquisition:** Anders Sönnerborg, J. Peter Svensson.

**Investigation:** Birgitta Lindqvist, Bianca B. Jütte, Luca Love, Wlaa Assi, Tugsan Tezil, J. Peter
Svensson.

**Project administration:** J. Peter Svensson.

**Resources:** Eric Verdin, J. Peter Svensson.

**Software:** J. Peter Svensson.

**Supervision:** J. Peter Svensson.

**Writing – original draft:** J. Peter Svensson.

**Writing – review & editing:** Bianca B. Jütte, Luca Love, Wlaa Assi, Julie Roux, Anders
Sönnerborg.

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
