## [Decision Letter · Decision Letter 0]

25 Mar 2022

Dear Dr. Svensson,

Thank you very much for submitting your manuscript "T cell stimulation remodels the latently HIV-1 infected cell population by differential activation of proviral chromatin" for consideration at PLOS Pathogens. As with all papers reviewed by the journal, your manuscript was reviewed by members of the editorial board and by several independent reviewers. The reviewers appreciated the attention to an important topic. Based on the reviews, we are likely to accept this manuscript for publication, providing that you modify the manuscript according to the review recommendations. This may include additional experiments, as suggested by the 2nd reviewer, to ensure that the PLA method is appropriately controlled.

Sincerely,

Susan R. Ross, PhD

Section Editor

PLOS Pathogens

Susan Ross

Section Editor

PLOS Pathogens

Kasturi Haldar

Editor-in-Chief

PLOS Pathogens

orcid.org/0000-0001-5065-158X

Michael Malim

Editor-in-Chief

PLOS Pathogens

orcid.org/0000-0002-7699-2064

Reviewer Comments (if any, and for reference):

Reviewer's Responses to Questions

**Part I - Summary**

Reviewer #1: This study presents a new approach to understanding molecular events at the HIV LTR during reactivation of latent proviruses. The revised study addresses most of the concerns raised in the initial review.

Reviewer #2: The authors have developed a J-Lat based cell system to use as a model for latency reversal. Their idea that Tat-ZFP3 PLA technique can selectively identify productive, Tat-dependent HIV-1 activation among non-physiological Tat independent activation is intriguing. However, their idea is not supported with experimental evidence – they use GFP accumulation as a control, but observe quiet significant discrepancies between Tat-ZFP3 and GFP.

**Part II – Major Issues: Key Experiments Required for Acceptance**

Reviewer #1: see above

Reviewer #2: The question about the specificity of the method has already been raised previously, and there (point 18 reviewer 1) the authors argued that the GFP expression occurs at different (later) time point with respect to Tat binding. This is indeed plausible, as shown also in Figure 3E, lines 257-259. However, while the levels of GFP continue to increase, Tat-ZFP3 PLA signals peak at 18hrs and then show a sharp drop. Therefore the authors opt for 16 hrs but the discrepancies remain, with their explanation being that the discrepancy between Tat-ZFP3 and GFP can be due to two mechanisms:

-transient association of Tat with the TAR early in latency reversal followed by GFP accumulation

-Tat-independent mechanisms leading to GFP expression

This is a problem, because this leaves the authors with no appropriate controls for the PLA assay.

The best way to control Tat induced transcriptional activity of the LTR promoter is to actually measure HIV-1 transcription by RT-PCR using different primers, for short (abortive) transcripts as well as long ones.

The other way to control their experiments if to measure the number of viral particles, as the 1C10 cell produce and bud off non-infectious viral particles after activations (lines 266-267).

Other minor points are listed bellow:

• The discrepancy between the microscopy and flow cytometry data (rows 175-177 of the Msc,) could be interpreted as sensitivity of the microscope based detection, but unlikely are due to compromised cellular viability, simply because dead cells will not adhere to the slides. As these are Jurkat cells, which need to be adhered to slides, dead cells will simply not attach to slides.

• It is clear already from data in Figure 1 (Figure 1E) and then later from Figure 3A and 3B that there are significant differences between Tat-ZFP3 PLA results and expression of GFP in the same cells. From the results presented in Figure 3, where different LRAs were used, it seems that GPF detection scores higher for the detection capacity of HIV-1 activation. Number of GFP + cells in PMA/i is almost 30% whereas there are less than 6% of PLATatZFP3+ cells. Also, when LRA activity was compared, it seems that the all the LRAs give some background PLA signal, while the number of real positive cells is low. Differences between LRAs are much better appreciated with GPF as a readout. How were these signals normalized? The authors need to add HIV-1 mRNA levels as a readout. They can also specifically measure the GPF signal as mRNA level

• Chromatin signatures of the PLA positive cells remain very puzzling. Considering the fact that in PLA + cells GFP intensity increase upon PMAi induction concurrently with all histone marks , both activating ie K27ac and K4me3, as well as repressing, K27me3 and K9me3 is further complicating the interpretation of these data.

**Part III – Minor Issues: Editorial and Data Presentation Modifications**

Reviewer #1: none

Reviewer #2: (No Response)

PLOS authors have the option to publish the peer review history of their article (what does this mean?). If published, this will include your full peer review and any attached files.

Reviewer #1: No

Reviewer #2: No

Figure Files:

Data Requirements:

Reproducibility:

References:

---

## [Editor Report · Decision Letter 1]

26 Apr 2022

Dear Dr. Svensson,

We are pleased to inform you that your manuscript 'T cell stimulation remodels the latently HIV-1 infected cell population by differential activation of proviral chromatin' has been provisionally accepted for publication in PLOS Pathogens.

Best regards,

Susan R. Ross, PhD

Section Editor

PLOS Pathogens

Susan Ross

Section Editor

PLOS Pathogens

Kasturi Haldar

Editor-in-Chief

PLOS Pathogens

orcid.org/0000-0001-5065-158X

Michael Malim

Editor-in-Chief

PLOS Pathogens

orcid.org/0000-0002-7699-2064
---

## [Editor Report · Acceptance letter]

2 Jun 2022

Dear Dr. Svensson,

We are delighted to inform you that your manuscript, "T cell stimulation remodels the latently HIV-1 infected cell population by differential activation of proviral chromatin," has been formally accepted for publication in PLOS Pathogens.

Best regards,

Kasturi Haldar

Editor-in-Chief

PLOS Pathogens

orcid.org/0000-0001-5065-158X

Michael Malim

Editor-in-Chief

PLOS Pathogens

orcid.org/0000-0002-7699-2064